# Polycyclic Phenol Derivatives from the Leaves of *Spermacoce latifolia* and Their Antibacterial and α-Glucosidase Inhibitory Activity

**DOI:** 10.3390/molecules27103334

**Published:** 2022-05-22

**Authors:** Shao-Bo Liu, Lei Zeng, Qiao-Lin Xu, Ying-Le Chen, Tao Lou, Shan-Xuan Zhang, Jian-Wen Tan

**Affiliations:** 1State Key Laboratory for Conservation and Utilization of Subtropical Agro-Bioresources/Guangdong Key Laboratory for Innovative Development and Utilization of Forest Plant Germplasm, College of Forestry and Landscape Architecture, South China Agricultural University, Guangzhou 510642, China; sbliu0511@126.com (S.-B.L.); ltao99@yeah.net (T.L.); sxzhang@163.com (S.-X.Z.); 2Guangdong Provincial Key Laboratory of Silviculture, Protection and Utilization, Guangdong Academy of Forestry, Guangzhou 510520, China; zenglei@sinogaf.cn (L.Z.); cyingl@sinogaf.cn (Y.-L.C.)

**Keywords:** *Spermacoce latifolia*, phenol derivative, antibacterial, anti-*MRSA*, *α*-glucosidase inhibitor

## Abstract

Three new polycyclic phenol derivatives, 2-acetyl-4-hydroxy-6H-furo [2,3-g]chromen-6-one (**1**), 2-(1′,2′-dihydroxypropan-2′-yl)-4-hydroxy-6H-furo [2,3-g][1]benzopyran-6-one (**2**) and 3,8,10-trihydroxy-4,9-dimethoxy-6H-benzo[c]chromen-6-one (**8**), along with seven known ones (**3**–**7**, **9** and **10**) were isolated for the first time from the leaves of *Spermacoce latifolia*. Their structures were determined by spectroscopic analysis and comparison with literature-reported data. These compounds were tested for their in vitro antibacterial activity against four Gram-(+) bacteria: *Staphyloccocus aureus* (*SA*), methicillin-resistant *Staphylococcus aureus* (*MRSA*), *Bacillus cereus* (*BC*), *Bacillus subtilis* (*BS*), and the Gram-(−) bacterium *Escherichia coli*. Compounds **1**, **2**, **5** and **8** showed antibacterial activity toward *SA*, *BC* and *BS* with MIC values ranging from 7.8 to 62.5 µg/mL, but they were inactive to *MRSA*. Compound **4** not only showed the best antibacterial activity against *SA*, *BC* and *BS*, but it further displayed significant antibacterial activity against *MRSA* (MIC 1.95 µg/mL) even stronger than vancomycin (MIC 3.9 µg/mL). No compounds showed inhibitory activity toward *E. coli*. Further bioassay indicated that compounds **1**, **4**, **5**, **6**, **8** and **9** showed in vitro *α*-glucosidase inhibitory activity, among which compound **9** displayed the best *α*-glucosidase inhibitory activity with IC_50_ value (0.026 mM) about 15-fold stronger than the reference compound acarbose (IC_50_ 0.408 mM). These results suggested that compounds **4**, **8** and **9** were potentially highly valuable compounds worthy of consideration to be further developed as an effective anti-*MRSA* agent or effective *α*-glucosidase inhibitors, respectively. In addition, the obtained data also supported that *S. latifolia* was rich in structurally diverse bioactive compounds worthy of further investigation, at least in searching for potential antibiotics and *α*-glucosidase inhibitors.

## 1. Introduction

The *Spermacoce* genus in the Rubiaceae family comprises 250–300 plant species that are widely distributed in tropical and subtropical zones around the world, including America, Europe, Africa, Australia and Asia [1]. To date, some species of this genus have been utilized in traditional or folk medicine to treat many human diseases such as malaria, digestive problems, fever, hemorrhage, urinary and respiratory infections, headache and skin diseases [1,2]. Previously, only a small number of plants in the *Spermacoce* genus were phytochemically studied, but by those studies more than 60 structurally diverse natural compounds, including bioactive iridoids, flavonoids, alkaloids, terpenoids and phenolic compounds, were discovered [3,4,5], which suggested that the plants of *Spermacoce* genus would potentially be rich resources for searching for a new group of bioactive natural products.

*Spermacoce latifolia* Aubl. (Synonym: *Borreria latifolia* (Aubl.) K. Schum.) is an annual herb native to Central and South America. For most of the people in many tropical and subtropical countries around the world this plant is more frequently known as an exotic invasive plant harmful to the local original plant biodiversity [6]. In China, *S. latifolia* was first introduced into Guangdong and some other places as military horse feed in 1937, but then it soon escaped into the wild and by now it has spread across a large area in coastal provinces and regions in southern China, including Guangdong, Guangxi, Hainan, Yunnan, Hong Kong, Taiwan, Fujian and Zhejiang [7]. It is noteworthy that the root juice of *S. latifolia* has long been used as a traditional remedy in Nepal to treat patients suffering from malaria [8,9], suggesting that potential bioactive compounds with pharmaceutical significance should exist in this plant species. However, in the past decades, *S. latifolia* was seldom phytochemically studied, and only a few iridoid glycoside and diterpenoid compounds were reported [10].

Bacterial infection has long been a serious threat to human public health. Although the discovery of antibiotics in the past century has led to a revolution in the treatment of bacterial infection diseases, the variation and drug resistance accumulation of pathogenic bacteria are significantly shortening the lifespan of clinically used antibiotics. Therefore, it is urgently necessary to develop new and effective antimicrobial agents to deal with the new threat from bacterial infections [11]. Diabetes mellitus type 2 (DM2) is another rapidly growing public health problem around the world, which is affecting more than 10% of the world population’s health [12,13]. An effective way to treat patients suffering DM2 is to orally use hypoglycemic drugs such as *α*-glucosidase inhibitors, which can control the blood glucose level by suppressing carbohydrate digestion [14]. Currently, some synthetic *α*-glucosidase inhibitors such as acarbose and voglibose are widely used as hypoglycemic agents in clinics to treat DM2, but they also cause various undesirable side effects, including flatulence, nausea and diarrhea [14,15]. Therefore, the search for more effective and safer naturally occurring *α*-glucosidase inhibitors has been drawing much more attention.

Previously, we had noticed in pre-test that the ethanol extract of *S. latifolia* had low toxicity and was capable of displaying antibacterial and *α*-glucosidase inhibitory potential. In addition, a recent phytochemical investigation on this plant had led to the isolation of some triterpenoids and anthraquinones with antibacterial or *α*-glucosidase inhibitory activity [16,17], which suggested that rich antibacterial and *α*-glucosidase inhibitory chemicals would exist in this plant. With the aim to further clarify those undiscovered antibacterial and *α*-glucosidase inhibitory compounds in this plant species, we initiated this phytochemical study on the leaves of *S. latifolia*, by which 10 polycyclic phenol derivatives, including 3 new (**1**, **2** and **8**) and 7 known ones (**3**–**7**, **9** and **10**) (Figure 1), were obtained. The structural establishment of the three new compounds were achieved by detailed analysis of their MS and NMR spectra (see Appendix A). Herein, we report the isolation and structure elucidation of these compounds, as well as the evaluation of their in vitro antibacterial and *α*-glucosidase inhibitory activity.

## 2. Results and Discussion

### 2.1. Structure Elucidation of the Compounds

The air-dried and powdered leaf material of *S. latifolia* was extracted with 95% ethanol, and the resultant crude ethanol extract was then sequentially partitioned with petroleum ether, ethyl acetate (EtOAc) and n-butanol, respectively. The petroleum ether-soluble and EtOAc-soluble fractions of the crude ethanol extract were subjected to a series of column chromatographic fractionation steps over silica gel, ODS and Sephadex LH-20 to afford the three new (**1**, **2** and **8**) and seven known (**3**–**7**, **9** and **10**) polycyclic phenol derivatives.

Compound **1** was obtained as a yellow powder with molecular formula C_13_H_8_O_5_ as determined by HR-ESI-MS, *m*/*z* 243.0300 [M−H]^−^ (calcd. for C_13_H_7_O_5_^−^ 243.0299), which requires 10 degrees of unsaturation. The ^1^H NMR spectrum of **1** (Table 1) showed signals readily recognized for a tertiary methyl group at *δ*_H_ 2.64 (3H, s, H-2′), two aromatic protons at *δ*_H_ 7.07 (1H, s, H-9) and 7.96 (1H, s, H-3), and two *cis*-oriented vicinal olefinic protons at *δ*_H_ 6.42 (1H, d, *J* = 9.6 Hz, H-7) and 7.96 (1H, d, *J* = 9.6 Hz, H-8), respectively. The ^13^C NMR and DEPT spectra (Table 1), coupled with HSQC analysis, of **1** supported the above analysis, which showed 13 carbon signals, including the presence of 1 methyl (*δ*_C_ 25.2), 1 carbonyl carbon (*δ*_C_ 188.4), 1 carboxyl carbon (*δ*_C_ 161.2), 4 olefinic methines and 6 olefinic quaternary carbons. These above findings supported **1** to be a highly conjugated compound bearing a basic coumarin skeleton. Careful analysis of the NMR data (Table 1) of **1** with those of the known compound xanthotoxol indicated that they were structurally closely related [18], with the only major difference being the signals for one additional acetyl group located at C-2 in **1** than in xanthotoxol. This assignment was consistent with the molecular formula of **1** and further well-supported by the following HMBC analysis. In the HMBC spectrum (Figure 2), significant correlations from *δ*_H_ 2.64 (Me-2′) to *δ*_C_ 153.4 (C-2) and 188.4 (C-1′), and from *δ*_H_ 7.96 (H-3) to *δ*_C_ 188.4 (C-1′), 147.2 (C-9a) and 142.5 (C-4) were observed, which supported the direct linkages of C-2 with C-1′ and C-3, of C-2′ with C-1′, and of C-3a with C-3, C-9a and C-4. The HMBC correlations from *δ*_H_ 7.07 (H-9) to *δ*_C_ 118.0 (C-3a), 140.7 (C-4a) and 144.9 (C-8) supported the connection of C-9a with C-9, and of C-8a with C-8, C-4a and C-9. The linkages of C-7 with C-6 and C-8 were supported by HMBC correlations of *δ*_H_ 7.96 (H-8) to *δ*_C_ 110.7 (C-9) and 140.7 (C-4a). In addition, the coupling constant of H-7 (*J*_7,8_ = 9.6 Hz) further evidenced the *cis*- configuration of the double bond between C-7 and C-8. Therefore, compound **1** was determined as 2-acetyl-4-hydroxy-6H-furo [2,3-g]chromen-6-one.

Compound **2** was obtained as a purple amorphous powder. Its HR-ESI-MS (negative mode) showed a pseudo-molecular ion peak at *m*/*z* 275.0561 [M − H]^−^, corresponding to a molecular formula of C_14_H_12_O_6_ (calcd. for C_14_H_11_O_6_^−^, 275.0561). The ^1^H NMR spectrum of **2** (Table 1) showed signals for a tertiary methyl group at *δ*_H_ 1.62 (3H, s, H-3′), a hydroxymethyl group at *δ*_H_ 3.81 (2H, s, H-2′), two singlet aromatic protons at *δ*_H_ 6.99 (1H, s, H-3) and 6.80 (1H, s, H-9), and two vicinal *cis*-positioned olefinic protons at *δ*_H_ 6.32 (1H, d, *J* = 9.5 Hz, H-7) and 7.92 (1H, d, *J* = 9.5 Hz, H-8). The ^13^C NMR and DEPT spectra (Table 1) of **2**, coupled with HSQC analysis, revealed 14 carbons, among which were a methyl group at *δ*_C_ 23.8 (C-3′), a hydroxymethyl group at *δ*_C_ 69.5 (C-2′), an oxygenated *sp^3^* quaternary carbon at *δ*_C_ 73.1 (C-1′), 4 olefinic methine carbons at *δ*_C_ 119.9 (C-3), 114.1 (C-7), 147.1 (C-8), 108.7 (C-9), an ester bond carboxyl carbon at *δ*_C_ 163.8 (C-6) and 6 olefinic quaternary carbons. These spectral data closely resembled the NMR data of **1**, which supported **2** to be also a coumarin derivative bearing a similar molecular skeleton to that of **1**. Comparison of the ^1^H and ^13^C NMR spectral data of **2** with those of **1** indicated the major difference that the signals for the acetyl group at C-2 in **1** were absent in **2**. Instead, proton and carbon signals for a propylene glycol group [*δ*_H_ 3.81 (2H, s, H-2′), 1.62 (3H, s, H-3′); *δ*_C_ 73.1 (C-1′), 69.5 (C-2′), 23.8 (C-3′)] were exhibited in the spectra of **2**. This observation supported that the structure of **2**, as shown in Figure 1, was close to **1** with the only difference of the acetyl group at C-2 in **1** being replaced by a propylene glycol group in **2**. This deduction was in accordance with the molecular formula of **2**, and further well-supported by 2D-NMR analysis. In the HMBC spectrum (Figure 2), significant correlations from *δ*_H_ 3.81 (H_2_-2′) to *δ*_C_ 100.9 (C-2) and 23.8 (C-1′), and from *δ*_H_ 6.99 (H-3) to *δ*_C_ 73.1 (C-1′) were observed, which supported the direct linkages of C-1′ with C-2, C-2′, C-3′, and of C-2 with C-3. The HMBC correlations from *δ*_H_ 6.99 (H-3) to *δ*_C_ 143.9 (C-4) and 148.9 (C-9a) supported the linkages of C-3a with C-3, C-4 and C-9a. The observation of HMBC correlations from *δ*_H_ 6.80 (H-9) to *δ*_C_ 119.9 (C-3a), 142.0 (C-4a) and 147.1 (C-8) supported the direct linkages of C-9 with C-9a, and of C-8a with C-4a, C-8 and C-9. The exhibition of significant HMBC correlations from 7.92 (H-8) to *δ*_C_ 143.9 (C-4a) and 163.8 (C-6) supported C-8 linked with C-8a, and C-7 linked with C-8 and C-6. In addition, the coupling constant between H-7 and H-8 (*J*_7,8_ = 9.5 Hz) supported the *cis*-configuration of the double bond between C-7 and C-8. Thus, compound **2** was elucidated as 2-(1′,2′-dihydroxypropan-2′-yl)-4-hydroxy-6H-furo [2,3-g][1] benzopyran-6-one.

Compound **8**, obtained as a yellowish powder, was determined to have a molecular formula C_15_H_12_O_7_ on the basis of HR-ESI-MS data (*m*/*z* 327.0467 [M + Na]^+^, calcd. for C_15_H_12_NaO_7_, 327.0475), which required 10 unsaturated degrees. The ^1^H NMR spectrum of **1** showed proton signals for two oxygenated methyl groups at *δ*
_H_ 3.94 (3H, MeO-4) and 3.97 (3H, MeO-9), two mutually coupled ortho aromatic hydrogen atoms at *δ*
_H_ 8.64 (1H, d, 9.1, H-1) and 6.82 (1H, d, 9.1, H-2), and a singlet aromatic hydrogen atom at *δ*
_H_ 7.38 (1H, s, H-7). The ^13^C-NMR spectra, coupled with HSQC analysis, showed the resonances for 15 carbons, including 1 carboxyl carbon, 2 methoxy carbons, 3 *sp^2^* methine carbons and 9 *sp^2^* quaternary C-atoms (Table 2). Taking these spectral data and the molecular formula into consideration, the existence of two aromatic rings and three hydroxyl groups could further be deduced. Careful analysis of the ^1^H- and ^13^C-NMR spectra indicated that **8** closely resembled the known compound 3,4,8,9,10-pentahydroxydibenzo[b,d]pyran-6-one [19], except that the signals for the two hydroxyl groups at C-4 and C-9 in 3,4,8,9,10-pentahydroxydibenzo[b,d]pyran-6-one were replaced by the resonances for two methoxy groups [*δ*_H_ 3.94 (3H, s, MeO-4) and 3.97 (3H, s, MeO-9)] in **1**, respectively. These above findings supported preliminarily establishing the molecular structure of **1** as shown in Figure 1, and this deduction was further well-supported by the following 1D and 2D NMR data. First, the singlet signal property of H-7 and the identical coupling constant of H-1 and H-2 (*J*_1,2_ = 9.1 Hz) supported the pentasubstituted and the ortho tetrasubstituted benzene rings, respectively. Second, in the HMBC spectrum, the observation of ^1^H–^13^C long-range correlations from *δ*_H_ 7.38 (1H, H-7) to *δ*_C_ 163.2 (C-6), 142.9 (C-9) and 117.9 (C-10a), and from *δ*_H_ 3.97 (3H, MeO-) to *δ*_C_ 142.9 (C-9) supported the location of a methoxy group at C-9, the linkages of C-8 with C-7 and C-9, and the linkages of C-6a with C-7, C-10a and C-6a (Figure 2). The HMBC correlations from *δ*_H_ 8.64 (H-1) to *δ*_C_ 117.9 (C-10a) and 145.4 (C-4a), and from *δ*_H_ 6.82 (H-2) to 112.9 (C-10b) supported the linkages of C-10b with C-1, C-10a and C-4a. The HMBC correlations from H-1 to *δ*_C_ 151.3 (C-3), from H-2 to *δ*_C_ 136.0 (C-4) and from *δ*_H_ 3.94 (3H, MeO-) to 136.0 (C-4) supported the location of a methoxy group at C-4 and further evinced the direct linkage of C-3 with C-2 and C-4, respectively. Therefore, the structure of compound **8** was unambiguously elucidated as 3,8,10-trihydroxy-4,9-dimethoxy-6H-benzo[c]chromen-6-one.

The seven known compounds were identified as 7*H*-furo [3,2-*g*][1]benzopyran-7-one (**3**) [20], 5,9-dihydroxy-8-methoxy-2,2-dimethyl-7-(3-methylbut-2-enyl)-2H,6H-pyrano [3,2-b]xanthen-6-one (**4**) [21], esculetin (**5**) [22], isoscopoletin (**6**) [23] and 7,8-dihydroxy-6-methoxycoumarin (**7**) [24], 6,7,4′-trihydroxyflavone (**9**) [25], 6,7,8,4′-tetramethoxyflavone (**10**) [26] by comparison of the spectroscopic data with those in literatures. They were all obtained from *S. latifolia* for the first time.

### 2.2. Antibacterial Activity Evaluation

All of the 10 isolated compounds were evaluated for their in vitro antibacterial activity against 5 bacterial strains, including 4 Gram-(+) bacteria *Staphyloccocus aureus* (*SA*), methicillin-resistant *Staphylococcus aureus* (*MRSA*), *Bacillus cereus* (*BC*) and *Bacillus subtilis* (*BS*), and one Gram-(−) bacteria, *Escherichia coli* (*EC*), using a method as described in the experimental section. The resulting MIC values of these compounds, as compared to reference compounds of kanamycin and vancomycin, are listed in Table 3. Compounds **1**, **2**, **5** and **8** were found to show moderate or weak antibacterial activity against three tested Gram-(+) bacteria of *SA*, *BC* and *BS* with MIC values ranging from 7.8 to 62.5 µg/mL, but they were inactive towards *MRSA*. Compound **4** was revealed to display the strongest antibacterial activity against all the four Gram-(+) bacteria (including *MRSA*) with MIC values 1.95 ~ 3.9 µg/mL, which were comparable to the two reference compounds of kanamycin and vancomycin. None of the test compounds displayed antibacterial activity against the Gram-(−) bacteria *E. coli* in this bioassay.

It is noteworthy that compound **4** was found to show significant antibacterial activity against *MRSA*. Compound **4** structurally belonged to the group of prenylated xanthones, and some known structurally similar compounds of this group were reported to show several biological activities, including cytotoxic, antimicrobial, etc. [27]. *MRSA* infection is responsible for a rapidly increasing number of serious infectious diseases severely threatening global public health [28,29], and it has surpassed hepatitis B and AIDS, ranking first among the three most intractable infectious diseases throughout the world [30]. Although several antibiotics such as vancomycin, teicoplanin and daptomycin had been recommended for the treatment of *MRSA* infections [31,32], they showed a series of drawbacks such as slow bactericidal activity, low tissue penetration and increasing reports of resistance, which greatly restricted their clinical utility [33,34,35,36,37]. Accordingly, *MRSA* infection is urgently lacking effective and safe antimicrobial agents for its control and therapy. The discovery of even stronger in vitro anti-*MRSA* activity of compound **4** (MIC 1.95 µg/mL) than the reference compound of vancomycin (MIC 3.9 µg/mL) suggests that this compound could be worthy of consideration to be developed as an effective anti-*MRSA* agent.

### 2.3. α-Glucosidase Inhibitory Activity Evaluation

α-Glucosidase inhibitors are capable of inhibiting the conversion of carbohydrates into small-intestine-absorbable monosaccharides, and therefore they have the potential for the treatment of diabetes mellitus type 2 (DM2) by controlling blood sugar levels [38,39]. In this study, compounds **1–10** were then further tested for their in vitro *α*-glucosidase inhibitory activity, with acarbose employed as a reference compound. The resulting data (Table 4) revealed that compounds **1**, **4**, **5**, **6**, **8** and **9** were biologically active to show in vitro *α*-glucosidase inhibitory activity with IC_50_ values ranging from 0.026 to 0.525 mM, which are close to or more potent than positive control of acarbose (IC_50_ 0.408 mM). In particular, compound **9** showed the best *α*-glucosidase inhibitory activity with an IC_50_ value of 0.026 mM, which is about 15-fold stronger than the reference compound. Compounds **4** and **8** also showed good *α*-glucosidase inhibitory activity, with IC_50_ values more than 2-fold lower than acarbose. In addition, a comparison of the chemical structures and the activities of **9** and **10** indicated that the existence of free hydroxy groups at C-7 and C-4′ would be essential for this type of flavonoids to display *α*-glucosidase inhibitory activity. A negative effect on the *α*-glucosidase inhibitory activity of coumarin derivatives was also evident when a free hydroxy group was located at C-8 of the basic coumarin skeleton, as supported by comparison of the structures and activities of compounds **5**, **6** and **7**. Generally, the bioassay data not only indicated that rich phenolic derivatives with *α*-glucosidase inhibitory activity were existing in this plant, but also revealed that compounds **4**, **8** and **9** were potentially highly valuable *α*-glucosidase inhibitors worthy to be further developed as effective hypoglycemic agents for the treatment of DM2 patients [14].

As a successful invasive plant, *S. latifolia* normally grows very fast, and annually it can produce a huge plant biomass at its invasion habitats. However, due to the lack of essential study to discover its potential values, the abundant biomass resources of this plant have not been well-developed so far. In a recently conducted phytochemical investigation, nine ursane and oleanane triterpenoids and five anthraquinone compounds were discovered from this plant species, and some of them were revealed to selectively show antibacterial, cytotoxic or *α*-glucosidase inhibitory activity [16,17]. This research further identified 10 structurally diverse bioactive polycyclic phenol derivatives from the leaves of *S. latifolia*, among which compounds **4**, **8** and **9** were addressed with significant antibacterial and (or) *α*-glucosidase inhibitory activity. Collectively, the present study provided new data to support that *S. latifolia* is a plant worthy of further development in searching for structurally new and bioactive natural compounds. Moreover, compounds **4**, **8** and **9** were addressed as bioactive constituents of *S. latifolia* highly valuable to be further developed in medicinal or healthcare applications, at least as potential antibacterial agents or effective *α*-glucosidase inhibitors.

## 3. Materials and Methods

### 3.1. General Experimental Procedures

Optical rotations were determined on a Perkin-Elmer 341 polarimeter (Perkin-Elmer, Inc., Waltham, MA, USA). UV data were measured on a Perkin-Elmer Lambda 650 UV-Vis spectrophotometer (Perkin-Elmer, Inc., Waltham, MA, USA). 1D and 2D Nuclear magnetic resonance (NMR) spectra were recorded on a Bruker DRX-500 NMR spectrometer (Bruker Biospin Gmbh, Rheistetten, Germany) or on a Bruker advance 600 NMR spectrometer (Bruker, Karlsruhe, Germany), with tetramethylsilane (TMS) as an internal standard. Electrospray ionization-mass spectrometry (ESI-MS) were acquired on a *MDS SCIEX API 2000* liquid chromatography/tandem mass spectrometry (LC/MS/MS) instrument (Applied Biosystems, Inc., Forster, CA, USA). High-resolution (HR) ESI-MS data were obtained on an *API QSTAR Pulsar 1* spectrometer (Advanced Biomics, Los Angeles, CA, USA). Column chromatography (CC) was performed with silica gel (80–100 and 200–300 mesh, Qingdao Haiyang Chemical Co., Qingdao, China), Sephadex LH-20 (Pharmacia Fine Chemical Co. Ltd., Uppsala, Sweden), MCI gel CHP 20P (75–150 μM, Mitsubishi Chemical Corp., Tokyo, Japan). Medium pressure liquid chromatography (MPLC) was carried out on a *CXTH P3000* instrument (Beijing ChuangXinTongHeng Science and Technology Co., Ltd., Beijing, China) equipped with a *UV3000 UV-Vis* Detector and a Fuji-C18 column (400 mm × 25 mm i.d., 50 mM, YMC Co., Ltd., Kyoto, Japan). Thin-layer chromatography (TLC) was conducted on precoated silica gel plates (HSGF254, Yantai Jiangyou Silica Gel Development Co., Ltd., Yantai, China), and spot detection was performed by spraying 10% H_2_SO_4_ in ethanol, followed by heating. CD_3_OD, Kanamycin sulfate, Vancomycin, resazurin and *α*-glucosidase were purchased from Sigma Chemical Co. (Sigma-Aldrich Company, St. Louis, MO, USA). *p*-Nitrophenyl-*α*-d-glucopyranoside (PNPG) and acarbose were obtained from Tokyo Chemical Industry Co., Ltd. (Tokyo, Japan). Other solvents (analytical grade) including methanol, ethyl acetate, chloroform, n-butyl alcohol and acetone, were purchased from Tianjin Fuyu Fine Chemical Industry Co. (Tianjin, China). 

### 3.2. Plant Material

The leaf material of *S. latifolia* was collected around the campus of the South China Agricultural University, Guangzhou, China, in September 2019, identified by Dr. Hong-Feng Chen at South China Botanical Garden, the Chinese Academy of Sciences (CAS). A voucher specimen (No. 20190925) was deposited at the Laboratory of Phytochemistry at the College of Forestry and Landscape Architecture, South China Agricultural University.

### 3.3. Extraction and Isolation

Air-dried leaf material (13.5 kg) of *S. latifolia* was mechanically powdered at room temperature by a grinder. Then the powdered material was extracted three times (2 days each) with 95% EtOH (13.0 L × 3) at room temperature. The collected extraction solution was then evaporated under reduced pressure by a 20L-type Buchi evaporator to provide a black residue, which was then suspended in H_2_O (4.5 L) and successively partitioned with petroleum ether (4.5 L × 3), ethyl acetate (4.5 L × 3) and *n*-BuOH (4.5 L × 3) to afford petroleum ether-soluble (1.1 kg), EtOAc-soluble (610 g) and *n*-BuOH-soluble (570 g) fractions after condensation to dryness in vacuo.

Proper amount of the petroleum ether-soluble fraction (850 g) was subjected to silica gel column chromatography (100 × 10 cm i.d.) using a gradient system of petroleum ether-acetone (100:0, 20:1, 10:1, 5:1, 2:1, 1:1, 0:100, *v*/*v*, each 3.0 L) to give nine fractions (E_1_–E_9_) after pooled according to their TLC profiles. The fraction E_3_, eluted with petroleum ether-acetone of 10:1, was first separated by a silica-gel column chromatography eluted with petroleum ether-acetone of 20:1 to provide sub-fractions E_3_-_1_~E_3_-_3_. Sub-fraction E_3_-_2_ was then purified by a Sephadex LH-20 column eluted with 1:4 (*v*/*v*) of CHCl_3_/MeOH to afford compound **8** (3.3 mg). Sub-fraction E_3_-_3_ was also purified by passing a Sephadex LH-20 column eluted with 1:4 (*v*/*v*) of CHCl_3_/MeOH to afford compound **4** (3.0 mg). The fraction E_7_ (27 g) were first passed through a MCI gel column (20 × 4.0 cm i.d.) for depigmentation. The resultant methanolic eluate (14.5 g) of E_7_ was then sequentially separated by MPLC eluted with a gradient of methanol in water (30:70, 40:60, 50:50, 60:40, 70:30, 80:20, 90:10, 100:0, *v*/*v*, each 1.0 L) to give fractions E_7-1_~E_7-16_. Fraction E_7-3_ (0.76 g), obtained from the elution of MeOH/H_2_O of 30:70, was separated by repeated Sephadex LH-20 column (155 × 1.3 cm i.d.) chromatography eluted with MeOH/CHCl_3_ (1:4, *v*/*v*) to obtain compounds **1** (3.3 mg) and **6** (19.5 mg). The Fraction E_7-4_ (0.7 g), obtained from the elution of MeOH/H_2_O of 40:60, was subjected to a silica gel CC using a gradient solvent system of CHCl_3_-MeOH (100:0, 100:1, 100:3, 20:1, 10:1, 0:100, *v*/*v*) to provide subfractions of E_7-4-1_~E_7-4-6_, and the fraction E_7-4-2_ was further separated by Sephadex LH-20 column (155 × 1.3 cm i.d.) chromatography eluted with MeOH/CHCl_3_ (4:1, *v*/*v*) to obtain compound **3** (3.2 mg).

Proper amount of the EtOAc-soluble fraction (560 g) was subjected to silica gel CC (100 × 10 cm i.d.) using a gradient of CHCl_3_-MeOH (100:1, 80:1, 50:1, 25:1, 20:1, 15:1, 10:1, 5:1, 2:1, 0:100, *v*/*v*, each 3.0 L) to give ten fractions (F_1_-F_10_). Fraction F_5_ (28.0 g) was separated by MPLC using elution solvent system of MeOH-H_2_O (30:70, 40:60, 50:50, 60:40, 70:30, 80:20, 90:10, 100:0, *v*/*v*, each 1.0 L) to yield 16 sub-fractions (F_5-1_-F_5-16_). Sub-fraction F_5-4_ (0.6 g), obtained from the elution of MeOH/H_2_O of 40:60, was first subjected to a silica CC eluted with 80:1 (*v*/*v*) of CHCl_3_/MeOH and then further purified by Sephadex LH-20 column (155 × 1.3 cm i.d.) chromatography eluted with CHCl_3_/MeOH of 1:4 (*v*/*v*) to afford compounds **10** (2.8 mg). Fraction F_6_ (27.6 g) was separated by MPLC using MeOH-H_2_O (30:70, 40:60, 50:50, 60:40, 70:30, 80:20, 90:10, 100:0, *v*/*v*, each 1.0 L) to yield 20 sub-fractions (F_6-1_–F_6-20_). Sub-fraction F_6-6_ (5.4 g), obtained from the elution of MeOH/H_2_O of 40:60, was firstly separated by silica CC eluted with 90:1 (*v*/*v*) of CHCl_3_/MeOH and then further purified by Sephadex LH-20 column (155 × 1.3 cm i.d.) chromatography eluted with MeOH to afford compounds **2** (4.2 mg). Sub-fraction F_6-7_ (0.9 g), obtained from the elution of MeOH/H_2_O of 50:50, was purified by repeated Sephadex LH-20 column (155 × 1.3 cm i.d.) chromatography eluted with CHCl_3_/MeOH (1:4, *v*/*v*) to obtain compounds **5** (11.5 mg) and **7** (16.2 mg). Sub-fraction F_6-9_ (0.6 g), obtained from the elution of MeOH/H_2_O of 60:40, was further purified by Sephadex LH-20 column chromatography eluted with CHCl_3_/MeOH (1:4, *v*/*v*) to obtain compounds **9** (3.6 mg).

### 3.4. Spectroscopic Data of Compounds **1**, **2** and **8**

*2-Acetyl-4-hydroxy-6H-furo [2,3-g]chromen-6-one* (**1**). Yellow amorphous powder; [α${]}_{D}^{26}$ +3.2 (*c* 0.2, MeOH); UV (MeOH) *λ*_max_ (log *ε*) 202 (0.93), 233 (0.58), 306 (0.28) nm; ESI-MS (+) *m*/*z* 245 [M + H]^+^; ESI-MS (−) *m*/*z* 243 [M − H]^−^, HR-ESI-MS (−) *m*/*z* 243.0300 [M − H]^−^ (calcd for C_13_H_7_O_5_, 243.0299). ^1^H NMR (500 MHz) and ^13^C NMR (125 MHz) in CD_3_OD, see Table 1.

*2-(1**′,2**′-Dihydroxypropan-2**′-yl)-4-hydroxy-6H-furo [2,3-g]*[1]*benzopyran-6-one* (**2**). Purple amorphous powder; [α${]}_{D}^{26}$ +2 (*c* 0.3, MeOH); UV (MeOH) *λ*_max_ (log *ε*) 220 (0.89), 254 (0.96), 306 (0.28) nm; ESI-MS (+) *m*/*z* 277 [M + H]^+^; ESI-MS (−) *m*/*z* 275 [M − H]^−^, HR-ESI-MS (−) *m*/*z* 275.0561 [M − H]^−^ (calcd for C_14_H_11_O_6_, 275.0561). ^1^H NMR (500 MHz) and ^13^C NMR (125 MHz) in CD_3_OD, see Table 1.

*3,8,10-trihydroxy-4,9-dimethoxy-6H-benzo[c]chromen-6-one* (**8**). [*α*]${}_{D}^{20}$ 0 (*c* 0.21, MeOH); Yellow powder; HR-ESI-MS (pos.) *m*/*z* 327.0467 (calcd for C_15_H_12_NaO_7_, 327.0475); ESI-MS (pos.) *m*/*z* 305 [M + H]^+^, 327 [M + Na]^+^; ESI-MS (neg.) *m*/*z* 303 [M − H]^–^; For ^1^H NMR (600 MHz, CD_3_OD) and ^13^C NMR (150 MHz, CD_3_OD) spectroscopic data, see Table 2.

### 3.5. Antibacterial Assay

The antibacterial assay was monitored in 96-well plates by using a method as described previously [40,41]. Briefly, 100 μL indicator solution (resazurin in sterile water, 100 μg/mL) was first placed into each of the sterility control wells (11th column) on the 96-well plates, and the indicator solution (about 7.5 mL, 100 μg/mL) was mixed with test bacteria (5 mL, 10^6^ cfu/mL, OD_600_ = 0.07) followed by transferring (100 μL, each) to growth control wells (12th column) and all test wells (1–10th column). Then, each of 100 μL of test compounds (1 mg/mL) in beef extract peptone medium, along with the positive control solutions (prepared by adding kanamycin sulfate and vancomycin instead of the samples) and the negative control solution (3% DMSO of beef extract peptone medium), were applied to the wells in the 1st column of the plates. Once all samples and controls were properly applied to the 1st column of wells on the plate, half of the homogenized content (100 μL solution) was then parallel-transferred from the 1st column wells to the 2nd column wells, and each subsequent well was treated similarly (doubling dilution) up to the 10th column, followed by discarding the last 100 μL aliquot. Finally, the plates were incubated at 37 °C for 5–6 h until the color of growth control change to pink. The lowest concentration for each test compound at which color change occurred was recorded as the MIC value of the test compound. In the bioassay, Gram-(+) bacterial strains of *Staphyloccocus aureus* (CMCC26003), methicillin-resistant *Staphylococcus aureus* (*MRSA*), *Bacillus cereus* (CMCC63302), *Bacillus subtilis* (CMCC63501), and Gram-(−) bacterial strain of *Escherichia coli* (CMCC44102) were used for the test, and they were all purchased from the Guangdong Institute of Microbiology (Guangzhou, China).

### 3.6. α-Glucosidase Inhibition Assay

The *a*-glucosidase inhibitory activity of the ten isolated compounds was determined in 96-well microtiter plates following the method as described previously [15,42]. In brief, *α*-glucosidase (20 μL, 0.5 U/mL) and various concentrations (500, 250, 125, 62.5, 31.2, 15.6, 7.8, 3.9 µg/mL) of tested compounds (120 μL) in 67 mM phosphate buffer (pH 6.8) were mixed in 96-well microtiter plates at room temperature for 10 min. Reactions were initiated by addition of 20 μL of 5.0 mM p-nitrophenyl-α-d-glucopyranoside (PNPG). The reaction mixture was incubated for 15 min at 37 °C in a final volume of 160 μL. Then, 0.2 M Na_2_CO_3_ (80 μL) was added to the incubation solution to stop the reaction and the absorbance was determined at 405 nm (for p-nitrophenol). The negative blank was set by adding phosphate buffer instead of test compounds via the same experimental procedure. Acarbose was utilized as positive control. The blank was set by adding phosphate buffer instead of the α-glucosidase using the same method. Inhibition rate (%) = [(ODnegative control − ODblank) − (ODtest − ODtest blank)]/(ODnegative blank − ODblank) × 100%. IC_50_ values were calculated and expressed as means ± standard deviations (SD) and SPSS 23.0 software was used for the analysis of variances.

## 4. Conclusions

Three new polycyclic phenol derivatives (**1**, **2** and **8**) along with seven known ones (**3**–**7**, **9** and **10**) were isolated from the leaves of *S. latifolia*. Their structures were determined by extensive spectroscopic analysis and comparison with literature-reported data. All the compounds were obtained from *S. latifolia* for the first time. Bioassays indicated that compounds **1**, **2**, **5** and **8** were active to show antibacterial activity toward three test Gram-(+) bacteria *SA*, *BC* and *BS*, but they were inactive to *MRSA*. Compound **4** was revealed to display the best antibacterial activity against all the four tested Gram-(+) bacteria (including *MRSA*) with MIC values comparable to reference compounds of kanamycin and vancomycin, and **4** in particular showed antibacterial activity against *MRSA* even stronger than vancomycin. No compounds showed inhibitory active toward the Gram-(−) bacteria *E. coli*. Further bioassay indicated that compounds **1**, **4**, **5**, **6**, **8** and **9** showed in vitro *α*-glucosidase inhibitory activity with IC_50_ values close to or more potent than acarbose (IC_50_ 0.408 mM), especially for compound **9** which displayed *α*-glucosidase inhibitory activity (IC_50_ 0.026 mM) about 15-fold stronger than the reference compound. The current research provided new data to support that *S. latifolia* is a plant rich in structurally diverse chemicals worthy of further development in medicinal or healthcare applications. In particular, compounds **4**, **8** and **9** were discovered to be the three most highly valuable compounds worthy to be developed as an effective anti-*MRSA* agent or effective *α*-glucosidase inhibitor, respectively.

## Figures and Tables

**Figure 1 molecules-27-03334-f001:**
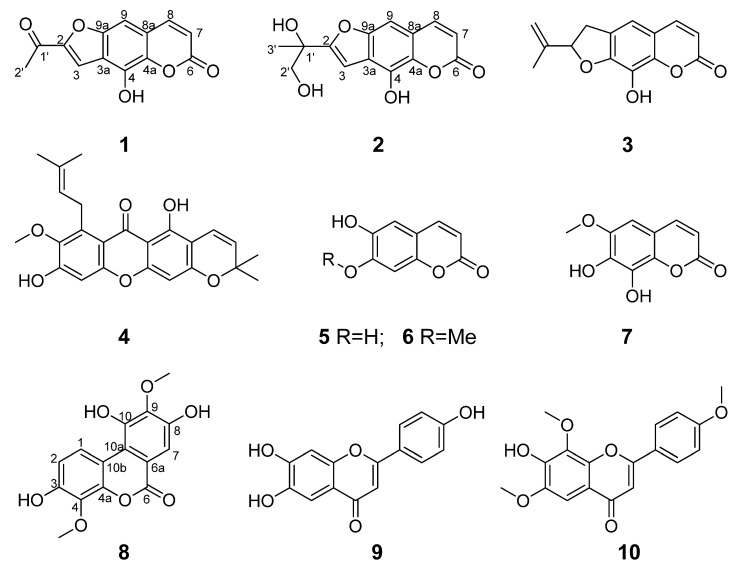
Chemical structures of compounds **1–10**.

**Figure 2 molecules-27-03334-f002:**
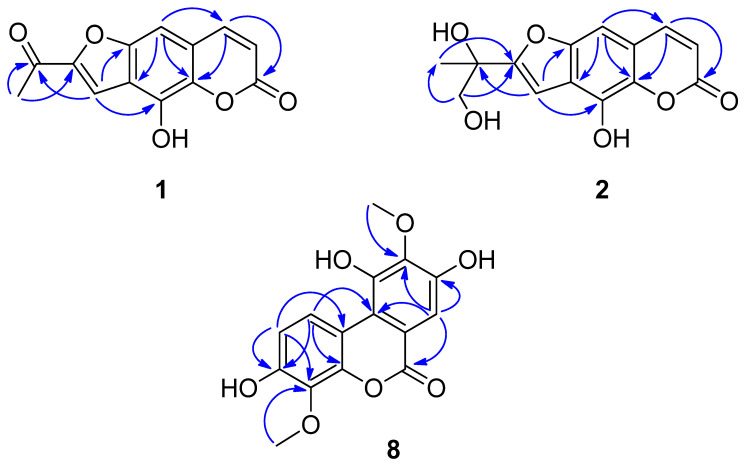
Key HMBC (

) correlations of new compounds **1**, **2** and **8**.

**Table 1 molecules-27-03334-t001:** ^1^H and ^13^C NMR NMR data of compounds **1** and **2**, *δ* in ppm and *J* in Hz.

No	*δ*_C_ (1)	*δ*_H_ (1)	*δ*_C_ (2)	*δ*_H_ (2)
2	153.4 C	―	164.0 C	―
3	110.1 CH	7.96 (s)	100.9 CH	6.99 (s)
3a	118.0 C	―	119.9 C	―
4	142.5 C	―	143.9 C	―
4a	140.7 C	―	142.0 C	―
6	161.2 C	―	163.8 C	―
7	114.2 CH	6.42 (d, 9.6)	114.1 CH	6.32 (d, 9.5)
8	144.9 CH	7.96 (d, 9.6)	147.1 CH	7.92 (d, 9.5)
8a	114.8 C	―	115.7 C	―
9	110.7 CH	7.07 (s)	108.7 CH	6.80 (s)
9a	147.2 C	―	148.9 C	―
1′	188.4 C	―	73.1 C	―
2′	25.2 CH_3_	2.64 (s)	69.5 CH_2_	3.81 (s)
3′			23.8 CH_3_	1.62 (s)

Data were measured at 500 MHz for ^1^H and 125 MHz for ^13^C NMR in CD_3_OD.

**Table 2 molecules-27-03334-t002:** ^1^H and ^13^C NMR data of compound **8**, *δ* in ppm and *J* in Hz.

No.	*δ* (H)	*δ* (C)	No.	*δ* (H)	*δ* (C)
1	8.64 (1H, d, 9.1)	123.7 t	8	−−	151.4 q
2	6.82 (1H, d, 9.1)	113.4 t	9	−−	142.9 q
3	−−	151.3 q	10	−−	149.3 q
4	−−	136.0 q	10a	−−	117.9 q
4a	−−	145.4 q	10b	−−	112.9 q
6	−−	163.2 q	4-OCH_3_	3.94 (3H, s)	61.7 s
6a	−−	116.6 q	9-OCH_3_	3.97 (3H, s)	61.0 s
7	7.38 (1H, s)	108.9 t			

Data were measured at 600 MHz for ^1^H and 150 MHz for ^13^C NMR in CD_3_OD.

**Table 3 molecules-27-03334-t003:** Antibacterial activity of compounds **1**–**10** (MIC, µg/mL).

Compounds	*SA*	*MRSA*	*BS*	*BC*	*EC*
**1**	62.5	>100	31.25	31.25	>100
**2**	62.5	>100	62.5	62.5	>100
**3**	>100	>100	>100	>100	>100
**4**	1.95	1.95	1.95	1.95	>100
**5**	31.25	>100	7.81	31.25	>100
**6**	>100	>100	>100	>100	>100
**7**	>100	>100	>100	>100	>100
**8**	15.6	>100	15.6	31.25	>100
**9**	>100	>100	>100	>100	>100
**10**	62.5	>100	15.6	62.5	>100
**K**	1.95	>100	1.95	3.9	1.95
**V**	1.95	3.9	1.95	1.95	>100

K = Kanamycin sulfate; V = Vancomycin.

**Table 4 molecules-27-03334-t004:** *α*-Glucosidase inhibitory activity of compounds **1**–**10**.

Compounds	IC_50_ (mM)	Compounds	IC_50_ (mM)
**1**	0.525 ± 0.004 ^a^	**6**	0.237 ± 0.001 ^d^
**2**	>1.0	**7**	>1.0
**3**	>1.0	**8**	0.184 ± 0.002 ^e^
**4**	0.162 ± 0.002 ^e^	**9**	0.026 ± 0.001 ^f^
**5**	0.284 ± 0.003 ^c^	**10**	>1.0
		Acarbose	0.408 ± 0.006 ^b^

Values represent mean ± SD (*n* = 3) based on three individual experiments. Different letters indicating significant differences labeled at the inhibitory activity at different compounds (*p* < 0.01).

## Data Availability

Not applicable.

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
