# Peer review of "Polycyclic Phenol Derivatives from the Leaves of Spermacoce latifolia and Their Antibacterial and α-Glucosidase Inhibitory Activity"

_molecules, 2022, doi:10.3390/molecules27103334_

Round 1

Reviewer 1 Report

In the paper entitled "Polycyclic phenol derivatives from the leaves of Spermacoce latifolia and their antibacterial and α-glucosidase inhibitory activity" the authors have presented 10 compounds obtained from the Spermacoce latifolia leaves and their antimicrobial activity as well as α-glucosidase inhibitory activity.

3 of the compounds were new, while 7 were known.

The manuscript presents some interest, but some issues need to be addressed.

The authors should also talk about the biocompatibility of the compounds. For example, the haemolysis test can be easily performed and gives important results.

The authors also need to discussed the results, and compare the data found with other data from the literature. For the 7 known compounds there were previous studies reporting their antimicrobial activity.

Author Response

Response to Reviewer 1

In the paper entitled "Polycyclic phenol derivatives from the leaves of Spermacoce latifolia and their antibacterial and α-glucosidase inhibitory activity" the authors have presented 10 compounds obtained from the Spermacoce latifolia leaves and their antimicrobial activity as well as α-glucosidase inhibitory activity.

3 of the compounds were new, while 7 were known.

The manuscript presents some interest, but some issues need to be addressed.

The authors should also talk about the biocompatibility of the compounds. For example, the haemolysis test can be easily performed and gives important results.

Our answer: Due to the amounts left are too small for most of the ten isolated compounds, we currently are not able to do and wholly accomplish more bioassay experiments. But we have managed to revise and improve the introduction section.

The authors also need to discussed the results, and compare the data found with other data from the literature. For the 7 known compounds there were previous studies reporting their antimicrobial activity.

Our response: Based on literature reports obtained, we revised the context and added some information we think necessary for part of the known compounds in the context.

Reviewer 2 Report

The aim of the research was to evaluate the antimicrobial properties and alpha-glucosidase inhibitory activity of compounds isolated from the leaves of Spermacoce latifolia. Modern analytical methods confirming the structure of the isolated compounds and standard methods of assessing their antibacterial properties and α-glucosidase inhibitor activity were used in the study. The topic of the work is not very interesting, because in vitro tests are of little practical importance. On the other hand, if natural remedies with antibacterial properties are sought, the research carried out at work will be of great importance.

Detail comments:

  1. In the introduction to the manuscript, there is no explanation for the need to assess the activity of α-glucosidase inhibitors. There is also a lack of information why this research was chosen such a raw material and why this research is innovative.
  2. table 3 - beginning of a sentence with a capital letter. The abbreviations used in this table require an explanation.
  3. The method of extracting compounds from dried and powdered plant material requires some details. How were the leaves dried? Was it a convection dryer or air-dried material? What was the air humidity and how long did the drying take? What device was used to crush such large amounts of dried material? What evaporator company was used to remove such large amounts of reagents? Under what pressure conditions did this happen?
  4. As the method of isolating the analyzed compounds is very extensive, I recommend adding a simple diagram showing how the appropriate fractions were obtained.
  5. Unfortunately, because I come from Europe, I am not able to read the information contained in description 3.4 - 分子式为, 分子式为.
  6. The study did not use the analysis of variance to compare the results. Only the standard deviation was given, which does not show any significant differences, especially in the results of α-glucosidase inhibitory activity and antibacterial activity. I also recommend adding which program was used to calculate the SD. There is also no description of the design of the experiment. The description of the methodology shows that the analyzes were performed three times, but there is no information on how many times the compounds were extracted and isolated.
  7. The innovativeness of applied research is demonstrated mainly by the conclusions at the end of the manuscript. It would also be important to indicate future directions of research, taking into account the obtained results. Which of the tested compounds had the best properties and its isolation would be used in natural medicine?
  8. The quote from the discussion of the results in the manuscript “Interestingly, S. latifolia was previously utilized as military horse feed, suggesting that chemicals isolated from this plant should be no or low toxic to mammals and human being”. I would be careful to make such a hypothesis. There are many plants that are safe for animals but have some undesirable effects in humans. The fact that the plant was used to feed horses does not translate directly to the fact that it is a plant that is safe for humans. More convincing is the fact that it was made into infusions.
  9. In the assessment of antimicrobial activity, there is no attempt to link the known chemical structure of the compounds (their functional groups) with the obtained results. Based on previous published studies, is there no correlation between the chemical structure of the tested compounds and their antimicrobial properties?

Author Response

Reviewer 2

The aim of the research was to evaluate the antimicrobial properties and alpha-glucosidase inhibitory activity of compounds isolated from the leaves of Spermacoce latifolia. Modern analytical methods confirming the structure of the isolated compounds and standard methods of assessing their antibacterial properties and α-glucosidase inhibitor activity were used in the study. The topic of the work is not very interesting, because in vitro tests are of little practical importance. On the other hand, if natural remedies with antibacterial properties are sought, the research carried out at work will be of great importance.

 Detail comments:

 In the introduction to the manuscript, there is no explanation for the need to assess the activity of α-glucosidase inhibitors. There is also a lack of information why this research was chosen such a raw material and why this research is innovative.

Our response: We have revised the introduction section to add the lack information including more clearly to display the justification for evaluating α-glucosidase inhibitory activity.

  1. table 3 - beginning of a sentence with a capital letter. The abbreviations used in this table require an explanation.

Our response: For table 3 - the beginning word ‘antibacterial’ is now revised to ‘Antibacterial’.

  1. The method of extracting compounds from dried and powdered plant material requires some details. How were the leaves dried? Was it a convection dryer or air-dried material? What was the air humidity and how long did the drying take? What device was used to crush such large amounts of dried material? What evaporator company was used to remove such large amounts of reagents? Under what pressure conditions did this happen?

Our response: Accordingly, we have revised the first paragraph in section ‘3.3 Extraction and isolation’ to clarify the answers for these questions.

  1. As the method of isolating the analyzed compounds is very extensive, I recommend adding a simple diagram showing how the appropriate fractions were obtained.

Our response: The word ‘habitats’ is revised to ‘habitat’, and the word ‘around’ is revised to ‘in’.

  1. Unfortunately, because I come from Europe, I am not able to read the information contained in description 3.4 - 分子式为, 分子式为.

Our response: We have now delete the redundant words ‘分子式为’ in description 3.4.

  1. The study did not use the analysis of variance to compare the results. Only the standard deviation was given, which does not show any significant differences, especially in the results of α-glucosidase inhibitory activity and antibacterial activity. I also recommend adding which program was used to calculate the SD. There is also no description of the design of the experiment. The description of the methodology shows that the analyzes were performed three times, but there is no information on how many times the compounds were extracted and isolated.

Our response: We have now labeled significant differences in Table 4 for the results of α-glucosidase inhibitory activity and added the description of the program used to calculate the SD.

  1. The innovativeness of applied research is demonstrated mainly by the conclusions at the end of the manuscript. It would also be important to indicate future directions of research, taking into account the obtained results. Which of the tested compounds had the best properties and its isolation would be used in natural medicine?

Our response: We have now revised the abstract and the conclusion section to add the information here suggested.

  1. The quote from the discussion of the results in the manuscript “Interestingly, S. latifolia was previously utilized as military horse feed, suggesting that chemicals isolated from this plant should be no or low toxic to mammals and human being”. I would be careful to make such a hypothesis. There are many plants that are safe for animals but have some undesirable effects in humans. The fact that the plant was used to feed horses does not translate directly to the fact that it is a plant that is safe for humans. More convincing is the fact that it was made into infusions.

Our response: In the revised version manuscript, we have deleted that sentence ‘Interestingly, … human being’.

  1. In the assessment of antimicrobial activity, there is no attempt to link the known chemical structure of the compounds (their functional groups) with the obtained results. Based on previous published studies, is there no correlation between the chemical structure of the tested compounds and their antimicrobial properties?

Our response: We have tried to add the structure-active relationship descriptions in the context.

Reviewer 3 Report

  • General comment: the paper needs a language revision in full.
  • Line 17. I suggest changing to: "… were isolated for the first time from…" and then delete " All the compounds were obtained from S. latifolia for the first time" in lines 18-19.
  • Line 17. What means "extensive"? Is this term really needed?
  • Line 20. Delete "of".
  • Line 21. Delete "of".
  • Line 22. Delete "three gram-(+) bacteria of"
  • Correct "ug/mL" to "µg/mL" along all the paper.
  • Line 26. Delete " the gram-(−) bacteria of"
  • Line 27. Delete "obviously".
  • What is the link between antibacterial and alfa-glucosidase inhibitory activities? If there is none, what is the importance of evaluating alfa-glucosidase inhibitory activity?
  • Abstract lacks a conclusion and mention the perspectives from the study.
  • Line 36-37. Delete "ethnomedicinally"
  • Lines 45-46. Delete "belonging to the genus of Spermacoce in the family of Rubiaceae"
  • The importance of the biological activities studied in the paper should be demonstrated/justified in "Introduction"
  • In addition, a background about triterpenoids and quinone compounds, even not very detailed, should be provided.
  • Lines 180 and 182. Delete "of". Revise all paper to correct this mistake. It repeats along the next paragraphs.
  • Evaluation of possible action mechanisms of the compounds on the bacteria will highly improve the paper. The same for experiments on toxicity (or mention to previously done on this aspect).
  • The discussion on alfa-glucosidase inhibitory property of compounds also needs to be improved.
  • Line 267. The correct is "leaf" or "leaves"
  • Were the isolation procedures based in any reference(s)?
  • Why antibacterial activity was evaluated in a short period (5-6h)?

Author Response

Reviewer 3

  • General comment: the paper needs a language revision in full.

Our response: We have again carefully checked and improved the language expression of the paper as much as we can.

  • Line 17. I suggest changing to: "… were isolated for the first time from…" and then delete " All the compounds were obtained from S. latifolia for the first time" in lines 18-19.

Our response: We have accordingly revised the sentences as suggested to “… were isolated for the first time from…” in line 17 and delete “All the compounds were obtained from S. latifolia for the first time” in lines 18-19.

  • Line 17. What means "extensive"? Is this term really needed?

Our response: The word ‘extensive’ is deleted.

  • Line 20. Delete "of".

Our response: The word ‘of’ in line 20 is deleted.

  • Line 21. Delete "of".

Our response: The word ‘of’ in line 21 is deleted.

  • Line 22. Delete "three gram-(+) bacteria of"

Our response: The words ‘three gram-(+) bacteria of’ in line 22 are deleted.

  • Correct "ug/mL" to "µg/mL" along all the paper.

Our response: We have corrected “ug/mL” to “µg/mL” along all the manuscript.

  • Line 26. Delete " the gram-(−) bacteria of"

Our response: The words ‘the gram-(−) bacteria of’ in line 26 are deleted.

  • Line 27. Delete "obviously".

Our response: The word ‘obviously’ in line 27 is deleted.

  • What is the link between antibacterial and alfa-glucosidase inhibitory activities? If there is none, what is the importance of evaluating alfa-glucosidase inhibitory activity?

Our response: We have added one paragraph in the introduction section to more clearly display the justification for evaluating α-glucosidase inhibitory activity.

  • Abstract lacks a conclusion and mention the perspectives from the study.

Our response: We have added a conclusion sentence in the abstract.

  • Line 36-37. Delete "ethnomedicinally"

Our response: The word ‘ethnomedicinally’ in line 36~37 is deleted.

  • Lines 45-46. Delete "belonging to the genus of Spermacoce in the family of Rubiaceae"

Our response: The words ‘belonging to the genus of Spermacoce in the family of Rubiaceae’ in line 45-46 are deleted.

  • The importance of the biological activities studied in the paper should be demonstrated/justified in "Introduction"

Our response: We have added one paragraph in the introduction section to more clearly display the justification for evaluating α-glucosidase inhibitory activity.

  • In addition, a background about triterpenoids and quinone compounds, even not very detailed, should be provided.

Our response: We have revised the correlated sentences in the context to provide more information about the triterpenoids and quinone compounds isolated from S. latifolia.

  • Lines 180 and 182. Delete "of". Revise all paper to correct this mistake. It repeats along the next paragraphs.

Our response: The word ‘of’ in line 180 and 181, and along similarly in the next paragraphs are deleted.

  • Evaluation of possible action mechanisms of the compounds on the bacteria will highly improve the paper. The same for experiments on toxicity (or mention to previously done on this aspect).

Our answer: We currently have yet not accomplished the evaluation of possible action mechanisms of the compounds on the bacteria, due to the limitation of the amounts of the isolated compounds.

  • The discussion on alfa-glucosidase inhibitory property of compounds also needs to be improved.

Our response: Based on this comment, we have now revised and optimized the discussion on α-glucosidase inhibitory activity.

  • Line 267. The correct is "leaf" or "leaves"

Our response: The word ‘leave’ in line 267 is corrected to ‘leaf’.

  • Were the isolation procedures based in any reference(s)?

Our answer: The isolation procedures applied in this paper are designed on the principle of general technical methods of phytochemistry, rather than specifically based on a specialized method in a reference paper.

  • Why antibacterial activity was evaluated in a short period (5-6h)?

Our response: The bioassay method we used in this paper is to test the MIC values of tested compounds by 96-well plates, which is a standard method used in references [Phytochemistry, 2005, 66, 1601-1606; Molecules, 2014, 19, 4301-4312]. In this method, 5-6h is already sufficient in the control well for live bacteria to change the blue color of the indicator solution (resazurin in H2O) into pink.

Round 2

Reviewer 1 Report

The authors have made changes that improved the quality of the paper. Can be accepted for publication in the present form

Author Response

Reviewer 1

The authors have made changes that improved the quality of the paper. Can be accepted for publication in the present form

Our answer: Many thanks for this comment.

Reviewer 2 Report

Most of my comments have been incorporated in the revised manuscript. I have reservations about the statistical analysis presented in Table 4. Why are the results for α-glucosidase inhibitory activity of several compounds with IC 50> 1.0 described as belonging to different homogeneous groups?

Author Response

Reviewer 2

Most of my comments have been incorporated in the revised manuscript. I have reservations about the statistical analysis presented in Table 4. Why are the results for α-glucosidase inhibitory activity of several compounds with IC 50> 1.0 described as belonging to different homogeneous groups?

Our response: That is because that the exact α-glucosidase inhibitory activity (original IC50 data) of those compounds with IC50> 1.0 are different from each other. Considering that the exact data of the IC50 values for those compounds with IC50> 1.0 were not detailedly presented in Table 4, to indicate the differences between these compounds with IC50> 1.0 do not means more. Therefore, in the revised version manuscript, we delete to label the differences for those compounds in Table 4 with IC50> 1.0.

Reviewer 3 Report

The paper was highly improved after the review. However, I still have some suggestions for the authors?

  • Line 21. Correct to "gram(-) bacterium Escherichia coli"
  • Line 32. Delete "presently"
  • Line 65. Delete "in modern medicine"
  • Line 73. Replace "due to that they" by "which"
  • Introduction was highly improved and now the relevance of the study and its background are clear.
  • Line 222. Change "antimicrobe" to "antimicrobial"
  • Line 252. Replace "stronger" by "lower"
  • Lines 402-403. Why compounds concentrations are mentioned here in µg/mL but the IC50 values are expressed in mM? The same units should be used.

Author Response

Reviewer 3

The paper was highly improved after the review. However, I still have some suggestions for the authors?

  • Line 21. Correct to "gram(-) bacteriumEscherichia coli"

Our response: We have corrected the word ‘bacteria’ to ‘bacterium’.

  • Line 32. Delete "presently"

Our response: The word ‘presently’ has now been deleted.

  • Line 65. Delete "in modern medicine"

Our response: The words ‘in modern medicine’ have now been deleted.

  • Line 73. Replace "due to that they" by "which"

Our response: The words ‘due to that they’ is replaced by ‘which’.

  • Introduction was highly improved and now the relevance of the study and its background are clear.

Our response: Thanks for this comment.

  • Line 222. Change "antimicrobe" to "antimicrobial"

Our response: The word ‘antimicrobe’ is replaced by ‘antimicrobial’.

  • Line 252. Replace "stronger" by "lower"

Our response: The word ‘stronger’ is replaced by ‘lower’.

  • Lines 402-403. Why compounds concentrations are mentioned here in µg/mL but the IC50 values are expressed in mM? The same units should be used.

Our response: In the bioassay, it is more convenient to prepare the solution of tested compounds in µg/mL for the experiment. But, because different compounds have different molecular weight, it is not a better way to differentiate the α-glucosidase inhibitory activity of different compounds just by comparing the difference of their IC50 values expressed in µg/mL. Therefore, as also suggested by other experts, we expressed the IC50 values of the α-glucosidase inhibitory activity in molar concentration (mM).
